# The neural dynamics associated with computational complexity

**Juan Pablo Franco** [1], **Peter Bossaerts** [1,2], **Carsten Murawski** [1] *

**1** Centre for Brain, Mind and Markets The University of Melbourne, Melbourne, Victoria, Australia, **2** Faculty of Economics, Cambridge University, Cambridge, United Kingdom

* carstenm@unimelb.edu.au

**Data Availability Statement:** The data analysis code and the behavioral data are available at the OpenScience Framework (OSF: https://osf.io/g4h7y/). The anonymized neuroimaging data are available (in BIDS format) at OpenNeuro (https://

## Abstract

Many everyday tasks require people to solve computationally complex problems. However, little is known about the effects of computational hardness on the neural processes associated with solving such problems. Here, we draw on computational complexity theory to address this issue. We performed an experiment in which participants solved several instances of the 0-1 knapsack problem, a combinatorial optimization problem, while undergoing ultra-high field (7T) functional magnetic resonance imaging (fMRI). Instances varied in computational hardness. We characterize a network of brain regions whose activation was correlated with computational complexity, including the anterior insula, dorsal anterior cingulate cortex and the intra-parietal sulcus/angular gyrus. Activation and connectivity changed dynamically as a function of complexity, in line with theoretical computational requirements. Overall, our results suggest that computational complexity theory provides a suitable framework to study the effects of computational hardness on the neural processes associated with solving complex cognitive tasks.

## Author summary

Humans are frequently faced with complex decisions, ranging from everyday tasks like grocery shopping to more intricate decisions such as selecting an investment portfolio. These decisions require higher-order problem-solving skills, which remain poorly understood, particularly regarding the neural processes that support deliberation. In this study, we introduce a framework that employs computational complexity theory to investigate the neural activity that occurs during complex problem-solving. We suggest that the inherent characteristics of a problem determine its computational difficulty, and that these intrinsic features could be used to identify consistent neural patterns during complex problem-solving. To test this approach, we applied it to the knapsack problem, a standard computational problem. Participants solved several versions of this problem while their brain activity was monitored using ultra-high field MRI. By leveraging computational complexity theory, we developed generic metrics of computational difficulty and successfully identified the corresponding neural correlates and their dynamics during problem-solving. The results indicate that the proposed framework, grounded in computational complexity theory, offers a promising method for studying the neural processes

doi.org/10.18112/openneuro.ds005427.v1.1.1)
and through the OSF project.

**Funding:** JPF was supported by a University of Melbourne Graduate Research Scholarship from the Faculty of Business and Economics (https://fbe.unimelb.edu.au). PB acknowledges financial support through a R@MAP Chair from the University of Melbourne (https://unimelb.edu.au). The funders had no role in study design, data collection and analysis, decision to publish, or preparation of the manuscript.

**Competing interests:** The authors have declared that no competing interests exist.

involved in complex problem-solving. This approach could provide valuable insights into a topic that has previously resisted formal investigation.

## 1 Introduction

Every day, people make decisions that require solving complex problems. Many of these problems are known to be computationally intractable in the sense that the number of operations that need to be performed to find a solution grows quickly to levels that make solving these problems correctly infeasible. Real-life examples of intractable tasks include attention gating, task scheduling, shopping, routing, bin packing, and gameplay [1, 2]. Despite the relevance of intractable problems in daily life, little is known about the effects of complexity of tasks on the neural processes during problem-solving.

Intractable problems require an extended period of time to solve and involve an extensive search space. These two characteristics defy formal investigation of neural dynamics. Firstly, since solving such tasks requires an extended period of time, one cannot opt for modeling based on discrete choice theories such as those underlying neuroeconomics [3]. When deciding between, say, an apple and a candy, the neural activation can be modeled in terms of an indicator variable whose level is modulated by the value inferred from choices and kept constant throughout the short (couple of seconds) deliberation time [e.g., 4]. When choice concerns complex alternatives, deliberation times may be an order of magnitude longer, so neural activation can be expected to fluctuate markedly during the course of deliberation. Secondly, because the search space is large, there are a plethora of paths that can be chosen during resolution. Since human approaches to solving a complex problem exhibit substantial heterogeneity, both across individuals and over time [e.g., 5], modeling neural dynamics during deliberation is bound to be challenging if it is to be based on "what people are thinking," i.e., on individual approaches to solving complex problems. A different strategy is called for.

Here, we propose to focus on "what people are solving," that is, on features of the computational tasks that are being presented. This has precedent in the analysis of probabilistic tasks, where intrinsic properties of the gamble at hand (such as mean and variance of the uncertain reward) have proven invaluable to deciphering the neural processes leading up to choice [e.g., 6]. Likewise, mathematical characteristics of the stimuli in perceptual tasks, such as signal strength, elucidate neural dynamics during deliberation [e.g., 7, 8]. Drawing on computational complexity theory, we demonstrate here that a mapping exists between intrinsic properties of instances of a problem related to computational hardness and neural dynamics during decision-making. Importantly, these properties represent generic features of computational problems that can be studied across different tasks and, indeed, have been shown to affect human behavior such as accuracy and time-on-task in several tasks [9, 10].

We studied the case of a canonical intractable (specifically "NP-complete"; see definition in Materials and methods) problem, the 0–1 knapsack decision problem (KP). There, the decision-maker is asked to choose whether, given a set of items with differing value and weight, there exists a subset whose total value is at least as high as a given threshold, while the total weight is less than or equal to a capacity constraint. We identified two properties of instances of KP related to an instance's computational hardness and tested whether these properties elucidated neural signatures during deliberation. The two properties are *complexity* and *proof hardness*. Complexity captures the number of computational steps (or time) needed to solve an instance, while proof hardness represents the computational steps needed to verify the correctness of the solution.

In order to measure complexity, we utilized a metric of difficulty that arises from the study of *random ensembles of instances* (i.e., random cases of the problem). Variation in expected computational complexity, regardless of the algorithm used, has been attributed to specific structural properties of instances [11–17]. The resulting "typical-case complexity" (TCC) has been found to affect human performance and effort in several intractable (NP-complete) problem-solving tasks, including KP [9, 10]. Therefore, we hypothesized that TCC would prove useful in characterizing the effects of computational hardness on neural processes. In analogy with work on neural correlates related to deliberation during tractable tasks [18–21], we expected neural correlates of TCC to overlap with the multiple-demand system. Specifically, we hypothesized they would overlap with two networks, (1) the cingulo-opercular network (CON), consisting of the dorsal anterior cingulate cortex (dACC) and the anterior insula (AI), and (2) the frontoparietal network (FPN), composed of the intraparietal sulcus (IPS) and specific regions from the lateral prefrontal cortex including the inferior frontal sulcus and the middle frontal gyrus (MFG) (e.g., [18, 21–23]). Additionally, we expected the level of complexity to be associated with neural markers of efficacy [24] and performance [25, 26].

We appeal to the theory of proof complexity to measure proof hardness. In the context of an NP-complete problem, such as the KP, there exists an asymmetry in the difficulty of proving that the solution is correct, which depends on the "satisfiability" of the instance. If an instance is *satisfiable* (the correct choice is 'yes'), it suffices to find a witness (example assignment of variables) that satisfies all of the constraints; one can then quickly verify that the witness indeed satisfies all the constraints, and this verification is tractable (i.e., can be done in polynomial time). For example, in the KP it suffices to find a subset of items that satisfies the weight and value constraints. In contrast, to confirm that an instance is *unsatisfiable* (the correct choice is 'no') requires proving that *no* witness exists, which is far more difficult (not tractable): even if a few potential witnesses are found not to satisfy the constraints, there may exist others that do. We thus conjectured that satisfiability would correlate with subjective *reliability*, that is, the degree to which the result of a calculation can be relied on to be accurate—much like variance modulates subjective beliefs of choice correctness in probabilistic tasks. Therefore, we expected neural correlates of this measure in regions that have been previously shown to encode uncertainty, specifically in CON [25–28].

In our experiment, participants were asked to solve several instances of the knapsack decision problem, while undergoing functional magnetic resonance imaging (fMRI). Instances were drawn randomly but in a way that systematically varied their TCC and their satisfiability. Critically, in order to more precisely localize and track neural signals during deliberation, we employed ultra-high field (7 Tesla) fMRI.

## 2 Results

Twenty participants (14 female, 5 male, 1 other; age range = 18–35 years, mean age = 26.6 years) took part in this study. Each participant was asked to solve 56 instances of the knapsack decision task while undergoing ultra-high field MRI brain scanning (Fig 1a). Instances varied in their computational complexity (TCC) and their satisfiability (2×2 balanced factorial design; see Materials and methods). Recall that the latter, satisfiability, captures proof hardness. Specifically, it encapsulates the asymmetry in the difficulty of proving that the solution is correct. Satisfiable instances are considered to have low proof hardness (verifying that a witness satisfies the constraints can be done in polynomial time) while unsatisfiable instances have a high proof hardness (verifying a proof of unsatisfiability might require more than polynomial time).

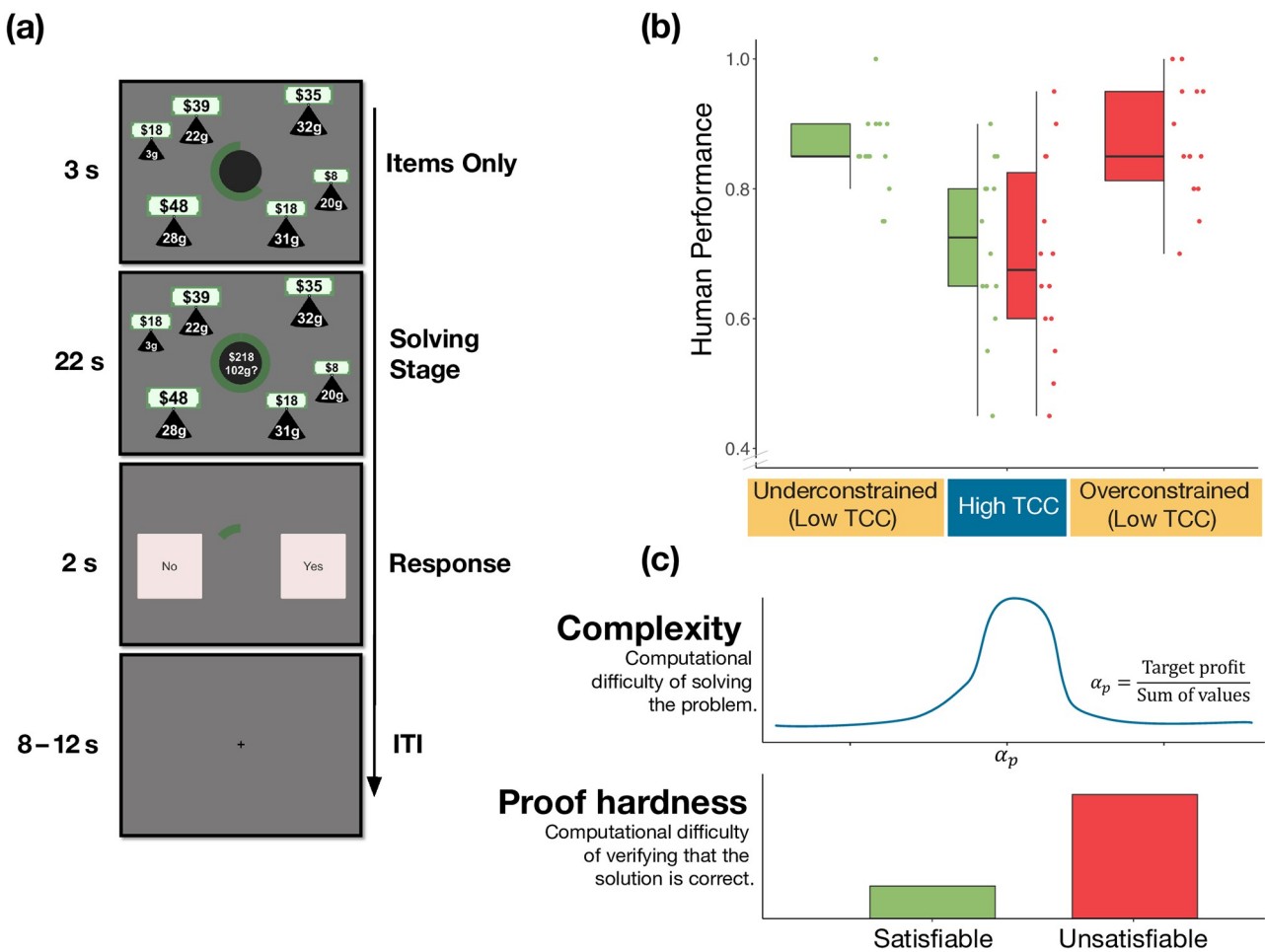

**Fig 1. (a) Task.** The task was composed of three main stages: the items stage (3 s), the solving stage (22 s) and the response stage (2 s). Initially, participants were presented with a set of items of different values and weights. The green circle at the center of the screen indicated the time remaining in this stage of the trial. This first stage lasted 3 seconds. Then, both capacity constraint and target profit were shown at the center of the screen. The objective of the task is to decide whether there exists a subset of items for which (1) the sum of weights is lower or equal to the capacity constraint and (2) the sum of values yields at least the target profit. This stage lasted 22 seconds. Finally, participants had 2 seconds to make either a 'YES' or 'NO' response using the response button box. A fixation cross was shown during the inter-trial interval (jittered between 8 and 12 seconds). **(b) Relation between TCC and human performance in the knapsack decision task.** Each dot represents an instance; human performance corresponds the proportion of participants that solved the instance correctly. Instances are categorized according to their constrainedness region ($\alpha_p$) and their TCC. In the underconstrained region (low TCC) the satisfiability probability is close to one, while in the overconstrained region (low TCC) the probability is close to zero. The region with high TCC corresponds to a region in which the probability is close to 0.5. Additionally, instances are categorized according to their solution (satisfiability) which is represented by their color. *The box plots represent the median, the interquartile range (IQR) and the whiskers extend to a maximum length of 1.5\*IQR.* **(c) Pictorial representation of complexity and proof hardness** as operationalized in relation to the properties of instances ($\alpha_p$ and satisfiability).

Additionally, participants performed, outside the scanner, a set of complementary tasks, including a knapsack optimization task and a set of cognitive function tasks. In this section, we report the behavioral results of the knapsack decision task, while the behavioral results from the complementary tasks are reported in Sections 3 and 5 in S1 Appendix.

## 2.1 Behavioral results

**2.1.1 Summary statistics.** On average, participants chose the 'YES' option in 50% of the trials (min = 25%, max = 68%). Mean *human performance*, measured as the proportion of trials

in which a correct response was made, was 0.78 (min = 0.48, max = 0.95, $SD$ = 0.14). Performance increased slightly as the task progressed; however, a negative (or null) effect cannot be fully ruled out with the evidence provided by the current dataset ($\beta_{0.5}$ = 0.009, $HDI_{0.95}$ = [−0.001, 0.021], main effect of trial number on performance, generalized logistic mixed model (GLMM); Table A Model 1 in S1 Appendix).

**2.1.2 Accuracy and instance properties.** We first studied the effect of TCC on human performance. This measure is based on a prominent framework in computer science that investigates the factors affecting computational hardness in computational problems by studying the difficulty of randomly generated instances of those problems. In the knapsack problem, TCC is explicitly connected to the normalized profit ($\alpha_p$) that captures the constrainedness of the problem. Explicitly, $\alpha_p$ is defined as the target profit (e.g., $218 in Fig 1a) divided by the sum of all item values in the instance. [9, 14]. This parameter captures the likelihood that a random instance is *satisfiable*, that is, that the solution is 'yes'. It can be used to specify regions where typical instances are generally satisfiable (under-constrained region), where they are unsatisfiable (over-constrained region), and where the probability of satisfiability is close to 50% (satisfiability threshold $\alpha_s$). As an illustration, consider the instance presented in Fig 1a. If the target profit were set to a large value (e.g., $300), the instance would be overconstrained (it is likely that no combination of items satisfies both constraints), whereas if it were set to a very low value (e.g., $20), the instance would be underconstrained. The more extreme these values are, the easier the instance is to solve. Indeed, it has been shown that the computational difficulty of solving the problem is higher when $\alpha_p$ is close to $\alpha_s$ [9, 14]. TCC is then defined based on the distance of $\alpha_p$ to the satisfiability threshold $\alpha_s$. Specifically, instances with values of $\alpha_p$ near the satisfiability threshold have a high typical-case complexity (*high TCC*) whereas instances further away from it—that is, in the under-constrained and over-constrained regions —have low typical-case complexity (*low TCC*). In line with previous results [9], we found participants performed better on instances with low TCC compared to those with high TCC ($\beta_{0.5}$ = −1.10, $HDI_{0.95}$ = [−1.44, −0.79], main effect of TCC on performance, GLMM; Fig 1b; Table A Model 2 in S1 Appendix).

We then studied proof-hardness by investigating satisfiability. Proof hardness is defined as the computational difficulty of verifying that the certificate of a solution (i.e., proof) is correct. An important driver of proof hardness is the satisfiability of an instance. To verify that an instance is satisfiable, it suffices to check that a candidate set of items (satisfiability certificate) satisfies the constraints. In contrast, verifying unsatisfiability requires validating a proof of non-existence (unsatisfiability certificate). For NP-complete problems, the former is tractable (P-time) whilst the latter is conjectured to be intractable (follows from the conjecture that $coNP \neq NP$; see Materials and methods).

Note that proof hardness is not directly related to the complexity of solving the knapsack problem. Instead, proof hardness characterizes the complexity of a different problem: the one of verifying that a solution to the problem is correct. Strictly speaking, it is mute to the complexity of finding the solution. As such, we hypothesized there would be no effect of satisfiability on performance. As expected, our findings replicate previous results that suggest that there is no effect of satisfiability on human performance in the knapsack decision task ([9]; $\beta_{0.5}$ = 0.02, $HDI_{0.95}$ = [−0.30, 0.30], main effect of satisfiability on performance, GLMM; Table A Model 5 in S1 Appendix). Moreover, we found no significant interaction effect between TCC and satisfiability on performance ($\beta_{0.5}$ = 0.26, $HDI_{0.95}$ = [−0.37, 0.90], interaction effect of TCC and satisfiability, GLMM; Fig 1b; Table A Model 6 in S1 Appendix). Besides studying and replicating previously reported effects of TCC and satisfiability on human performance, we replicated other key findings presented by Franco et al. [9] (see Section 3 in S1 Appendix).

Finally, we investigated human performance in a set of related tasks. We explored the relation between performance in the knapsack tasks and core cognitive abilities, including working memory, episodic memory, strategy use, as well as mental arithmetic. For this analysis, we utilized the joined data set from this study together with data collected by Franco et al. [9]. Our results suggest a weak relation between these cognitive abilities and performance in the knapsack tasks. The only significant correlation (at $\alpha = 0.05$) shows a link between mental arithmetic ability and performance in the knapsack optimization task (Section 5 in S1 Appendix).

## 2.2 Imaging results

**2.2.1 Whole-brain analysis.** We conducted a whole-brain analysis of the neural correlates of two intrinsic generic properties of problems: TCC and satisfiability. Additionally, we investigated the neural correlates of response accuracy (Section 6 in S1 Appendix). We did this by fitting GLMs that partitioned the solving stage into four separate periods (5.5 s) with an additional response stage modeled in the analysis (2 s).

**Neural correlates of TCC.** We expected to find activation related to complexity in regions in which activation had previously been shown to be correlated with cognitive demand (i.e., multiple-demand system). We explicitly expected to find evidence for the encoding of TCC in the cingulo-opercular network (CON) from early on during the solving stage due to its link with cognitive demand as well as its link with expected performance and reliability. Higher TCC entails, on average, lower performance and lower reliability of finding the solution (Fig 1b). Note that the estimation of TCC early on in the trial is feasible because constrainedness (and thus TCC) can be potentially estimated by performing sum and division operations, specifically, dividing the target profit by the sum of values.

We found that the neural correlates of TCC varied throughout the duration of the solving stage (Fig 2a, Table 1). Contrary to our expectations, we did not find significant correlations of TCC during the first period of the solving stage. Interestingly, during the second period, we did find a set of clusters that showed higher BOLD activity on instances with low TCC. These regions include the angular gyrus (AG) bilaterally, the superior frontal gyrus (SFG), the right middle frontal gyrus (MFG) as well as regions in the orbitofrontal cortex (bilaterally). It is worth noting that the negative pattern found in this period might stem from a different slope in the increased task-related activation and not from differences in the sustained level of activity (Fig 3). This pattern would align with previous results that support that the frontoparietal network (FPN) regions encode evidence accumulation towards a particular decision (see Discussion).

During the third period of the solving stage, the TCC contrast still showed significant clusters along the FPN, but the pattern overall changed, with respect to period S2. Critically, we found that a different set of regions within the FPN now showed a positive correlation with TCC. Specifically, we found positive clusters in the left SFG, left intraparietal sulcus (IPS), the cerebellum as well as a cluster in the right dorsolateral prefrontal cortex (dlPFC) in between the MFG and the SFG. Interestingly, the right AG kept on displaying a negative correlation with TCC during this period.

Finally, during the fourth, and last, period of the solving stage, a new set of clusters was identified. Markedly, this new set of clusters includes regions from both CON, FPN as well as significant clusters in the occipital lobe. In general, the activation in these clusters is correlated positively with TCC. The only two clusters that correlated negatively with TCC are those located in the ACC, as well as a cluster in the left SFG that overlaps with SFG cluster found in the second period.

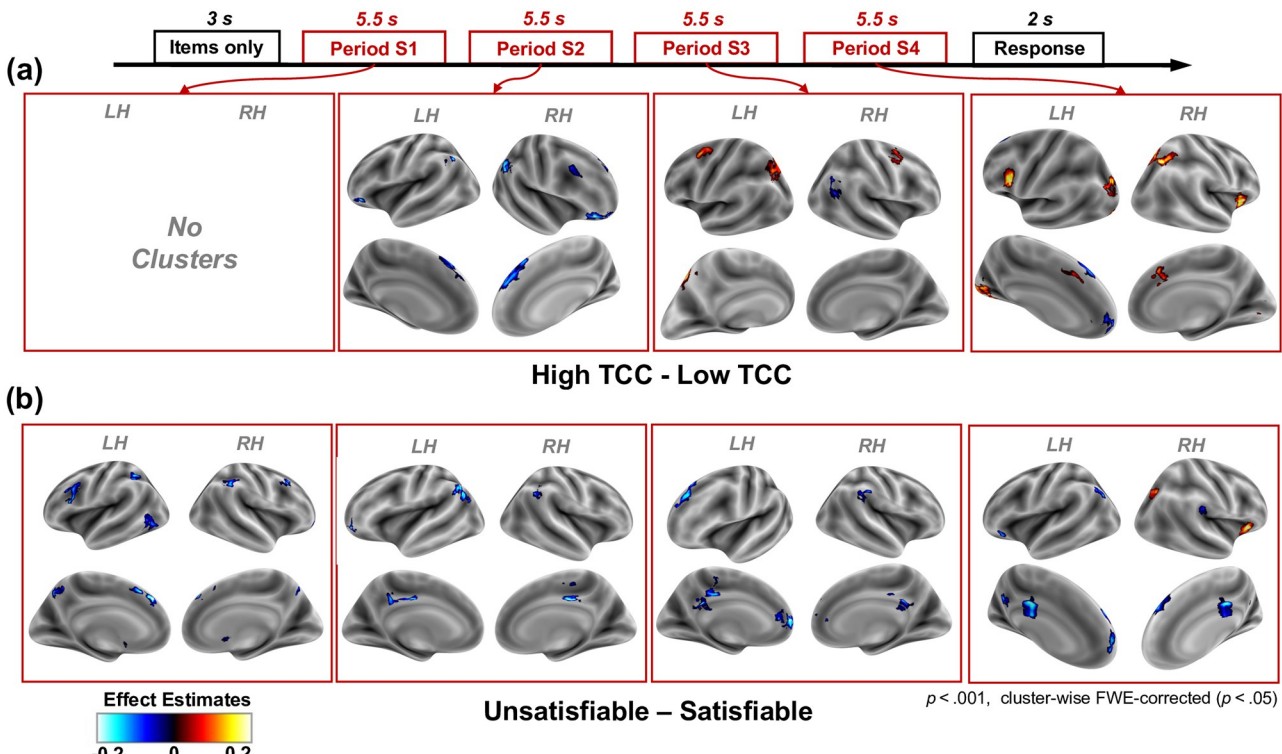

**Fig 2. Neural correlates of TCC and satisfiability. (a) Brain activation effect estimates ($\beta$) for the high vs. low TCC contrast ($\beta_{highTCC} - \beta_{lowTCC}$).** A positive contrast represents a higher BOLD signal for instances with high TCC compared to low TCC. Significant cluster-wise FWE-corrected ($p < 0.05$) clusters (with an uncorrected threshold of $p < 0.001$) are presented for each of the contrasts estimated using the Boxcar analysis. Each panel represents a different period in the solving stage. No significant clusters were found for period S1 nor for the response stage parameters. **(b) Brain activation effect estimates ($\beta$) for the unsatisfiable vs. satisfiable contrast ($\beta_{unsatisfiable} - \beta_{satisfiable}$).** A positive contrast represents a higher BOLD signal for unsatisfiable instances. Significant cluster-wise FWE-corrected ($p < 0.05$) clusters (with an uncorrected threshold of $p < 0.001$) are presented for each of the contrasts estimated using the Boxcar analysis. Each panel represents a different period in the solving stage. No significant clusters were found in the response stage.

Markedly, correlates of TCC on period S4 include the dACC and right anterior insula (AI) from CON as well as the precentral gyrus and IPS from FPN. The right IPS activation is segregated into two clusters, one medial and superior that overlaps with the precuneus and one more lateral that overlaps with the AG. These clusters were also found when using an alternative metric of complexity (instance complexity) that is closely related to TCC (see Sections 3 and 4 in S1 Appendix).

We did not find any significant clusters during the response stage.

**Neural correlates of satisfiability.** We expected the asymmetry between satisfiable and unsatisfiable instances to reflect differences in control signals associated with reliability (i.e., how much the result of a calculation can be relied on to be accurate). Specifically, we hypothesized that satisfiable instances would be associated with higher reliability, given that once a solution witness is found, verifying that the proposed solution is correct is a polynomial-time operation (tractable problem). In contrast, for unsatisfiable instances, verifying a proof of non-existence is conjectured to be intractable, and thus, computationally harder to verify.

Therefore, we expected regions that have been linked to monitoring of uncertainty to be more active during a trial with an unsatisfiable instance compared to a satisfiable one. In particular, we conjectured higher activation of the CON, on unsatisfiable instances, during late stages of the trial [25–28].

**Table 1. TCC clusters.** Significant cluster-wise FWE-corrected ($p < 0.05$) clusters (using an uncorrected threshold of $p < 0.001$) from the *High TCC—low TCC* contrast. Coordinates are in MNI space.

| Stage | Region* | Side | Cluster statistics | | | | Peak statistics | | |
|---|---|---|---|---|---|---|---|---|---|
| | | | Volume($mm^3$) | $\beta_{mean}$ | SEM | $\beta_{peak}$ | x | y | z |
| S2 | SFG | RH/LH | 4763.6 | -0.17 | 0.001 | -0.32 | 13 | 34 | 61 |
| | Orbitofrontal cortex | RH | 3878.9 | -0.21 | 0.002 | -0.37 | 51 | 44 | -19 |
| | AG | RH | 2662.4 | -0.20 | 0.002 | -0.31 | 51 | -55 | 37 |
| | AG | LH | 897.0 | -0.21 | 0.002 | -0.29 | -58 | -66 | 36 |
| | Orbitofrontal cortex | LH | 749.6 | -0.20 | 0.003 | -0.31 | -51 | 36 | -19 |
| | MFG | RH | 495.6 | -0.16 | 0.002 | -0.20 | 48 | 17 | 36 |
| S3 | IPS | LH | 2043.9 | 0.16 | 0.002 | 0.35 | -11 | -79 | 52 |
| | Cerebelum | RH | 938.0 | 0.11 | 0.002 | 0.19 | 0 | -60 | -25 |
| | SFG | LH | 786.4 | 0.14 | 0.002 | 0.19 | -26 | -2 | 52 |
| | AG | RH | 495.6 | -0.17 | 0.002 | -0.24 | 56 | -60 | 36 |
| | MFG/SFG | RH | 483.3 | 0.12 | 0.002 | 0.17 | 30 | -1 | 61 |
| S4 | Occipital Pole | LH | 2732.0 | 0.20 | 0.001 | 0.32 | -10 | -97 | -8 |
| | Fusiform gyrus | LH | 1888.3 | 0.19 | 0.002 | 0.31 | -24 | -79 | -14 |
| | Middle occipital gyrus | LH | 1503.2 | 0.20 | 0.002 | 0.28 | -29 | -78 | 21 |
| | AI | RH | 1265.7 | 0.21 | 0.003 | 0.30 | 32 | 28 | 0 |
| | Precentral gyrus | LH | 1163.3 | 0.22 | 0.003 | 0.33 | -43 | 4 | 24 |
| | IPS (precuneus) | RH | 1044.5 | 0.21 | 0.003 | 0.33 | 13 | -76 | 60 |
| | SFG | LH | 1024.0 | -0.15 | 0.003 | -0.27 | -16 | 36 | 58 |
| | dACC | LH/RH | 901.1 | 0.18 | 0.002 | 0.24 | -2 | 22 | 40 |
| | IPS (AG) | RH | 696.3 | 0.24 | 0.004 | 0.35 | 32 | -65 | 47 |
| | Occipital pole | LH | 667.6 | 0.26 | 0.003 | 0.34 | -38 | -95 | -6 |
| | ACC | LH | 475.1 | -0.22 | 0.003 | -0.29 | -5 | 57 | 8 |

*Region acronyms: SFG: superior frontal gyrus, AG: angular gyrus, MFG: middle frontal gyrus, IPS: intraparietal sulcus, AI: anterior insula, dACC: dorsal anterior cingulate cortex.

Interestingly, and contrary to our expectations, we found significant clusters from the first period of the solving stage (Fig 2b, Table 2). Moreover, significant clusters did not extend to the response screen, which was also in opposition to our conjecture. Most of the clusters during the solving stage showed a lower BOLD signal for unsatisfiable instances. These clusters extended from period S1 to period S4 of the solving stage. Notably, the posterior cingulate showed a lower sustained activation in unsatisfiable instances throughout the solving stage (periods S2, S3 and S4). Similarly, different clusters in the SFG had significant clusters throughout the solving stage. Additionally, similar to the clusters found for the TCC contrast, the AG showed bilateral activation during the second period of the solving stage. Interestingly, a bigger AG cluster was found on the left hemisphere compared to the right, in contrast to the right laterality predominance of AG found in the TCC contrast.

The only two clusters that showed significantly higher activity in unsatisfiable instances were the right AI and the occipital superior cortex, both present only during period S4 of the solving stage. The significant cluster found in the AI is in line with our hypothesis that unsatisfiable instances are related to higher markers of uncertainty, a signal that we expected to find in the CON. However, in disagreement with our hypothesis, we did not find a significant satisfiability cluster in the dACC. This may be due to insufficient statistical power of the whole brain analysis (see Fig 3b).

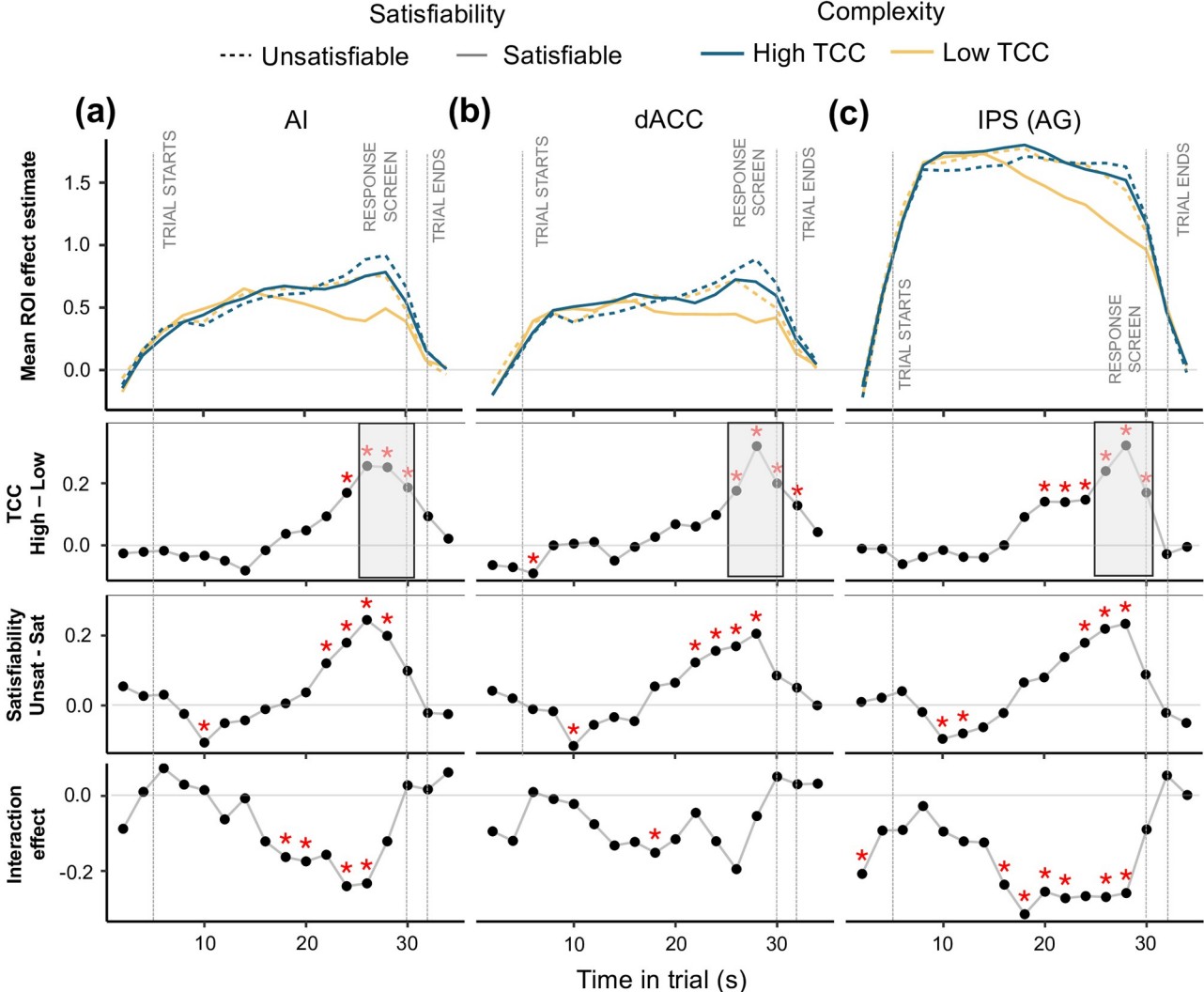

**Fig 3. Temporal dynamics of BOLD in regions of interest.** Mean effect estimate ($\beta$) for each ROI over time in trial. The effect at each time point represents the mean $\beta_{FIR}$ over all of the voxels from each ROI: right AI, dACC, and right IPS cluster extending to the angular gyrus. In the top row of panels, the $\beta_{FIR}$'s characterize the coefficients of an FIR regression with four conditions: satisfiability×TCC. The $\beta_{FIR}$ parameters are aligned to the BOLD signal, which has a lag with respect to the task time. To correct for this, the gray time-markers represent the task events by assuming a 5-second BOLD signal lag. In the second row, the TCC contrast ($\beta_{high} - \beta_{low}$) is presented. The third row displays the satisfiability contrast ($\beta_{unsat} - \beta_{sat}$). The bottom row shows the interaction effect between TCC and satisfiability ($[\beta_{highTCC,unsat} - \beta_{highTCC,sat}] - [\beta_{lowTCC,unsat} - \beta_{lowTCC,sat}]$). Red asterisks represent significance at the 0.05 level. Significance levels in the gray shaded regions are suggestive only; they represent the time period and contrast from which the ROIs were selected.

**Neural correlates of accuracy.** Although participants did not receive any feedback during the task, we expected to see error-related signals late in the trial. Although these signals would not represent the integration of novel exogenous information (since there was no feedback), we conjectured that participants would hold a subjective belief of the expected accuracy (or reward) of their answer (e.g, [29]). In line with our hypothesis, we found that activity in both FPN and CON was positively correlated with erring during the response stage (Section 6 in S1 Appendix).

**2.2.2 ROI dynamics.** Three ROIs were selected (see Section 4.9.3) to investigate more closely the neural dynamics associated with computational complexity. We included in our

**Table 2. Satisfiability clusters.** Significant cluster-wise FWE-corrected ($p < 0.05$) clusters (using an uncorrected threshold of $p < 0.001$) from the *Unsatisfiable-Satisfiable* contrast. Coordinates are in MNI space.

| Stage | Region | Side | Cluster statistics | | | | Peak statistics | | |
|---|---|---|---|---|---|---|---|---|---|
| | | | Volume($mm^3$) | $\beta_{mean}$ | SEM | $\beta_{peak}$ | x | y | z |
| S1 | SFG | LH | 1876.0 | -0.14 | 0.002 | -0.23 | -3 | 38 | 42 |
| | Supramarginal gyrus | RH | 1740.8 | -0.13 | 0.002 | -0.25 | 54 | -46 | 56 |
| | Supramarginal gyrus | LH | 1425.4 | -0.12 | 0.001 | -0.17 | -42 | -47 | 40 |
| | Inferior occipital cortex | LH | 1159.2 | -0.13 | 0.002 | -0.22 | -61 | -65 | -12 |
| | MFG | LH | 905.2 | -0.15 | 0.001 | -0.20 | -48 | 18 | 28 |
| | Caudate | LH | 880.6 | -0.17 | 0.002 | -0.24 | -8 | 2 | -1 |
| | MFG | RH | 868.4 | -0.12 | 0.003 | -0.23 | 37 | 30 | 53 |
| | Cerebellum | LH | 667.6 | -0.12 | 0.002 | -0.18 | -38 | -79 | -49 |
| | Precuneus | LH | 516.1 | -0.19 | 0.003 | -0.26 | -2 | -65 | 44 |
| | Frontal pole | RH | 450.6 | -0.14 | 0.002 | -0.18 | 29 | 58 | -9 |
| | Caudate | RH | 450.6 | -0.17 | 0.003 | -0.23 | 8 | 4 | 0 |
| S2 | AG | LH | 2523.1 | -0.19 | 0.002 | -0.29 | -62 | -60 | 29 |
| | Posterior cingulate | RH | 696.3 | -0.20 | 0.003 | -0.25 | 2 | -25 | 40 |
| | AG | RH | 585.7 | -0.18 | 0.003 | -0.28 | 62 | -57 | 34 |
| | Frontal pole | LH | 577.5 | -0.23 | 0.004 | -0.40 | -38 | 62 | -8 |
| | Posterior cingulate | LH | 479.2 | -0.16 | 0.002 | -0.19 | -10 | -41 | 37 |
| S3 | SFG | LH | 1511.4 | -0.15 | 0.002 | -0.22 | -21 | 36 | 55 |
| | Anterior cingulate | LH | 708.6 | -0.21 | 0.003 | -0.30 | -5 | 52 | 4 |
| | Supramarginal gyrus | RH | 696.3 | -0.14 | 0.002 | -0.20 | 64 | -28 | 39 |
| | Frontal pole | RH | 692.2 | -0.14 | 0.002 | -0.21 | 13 | 58 | 31 |
| | Anterior cingulate | LH | 593.9 | -0.15 | 0.002 | -0.24 | -6 | 46 | 12 |
| | Posterior cingulate | LH | 577.5 | -0.17 | 0.002 | -0.22 | -2 | -28 | 45 |
| | Posterior cingulate | LH | 487.4 | -0.18 | 0.004 | -0.28 | -2 | -44 | 28 |
| S4 | Posterior cingulate | RH | 1384.5 | -0.22 | 0.003 | -0.33 | 0 | -18 | 34 |
| | SFG | RH | 1306.6 | -0.18 | 0.002 | -0.26 | 14 | 50 | 42 |
| | AI | RH | 901.1 | 0.25 | 0.003 | 0.36 | 32 | 28 | 0 |
| | AG | LH | 659.5 | -0.24 | 0.003 | -0.30 | -48 | -68 | 44 |
| | SFG | LH | 647.2 | -0.17 | 0.004 | -0.26 | -14 | 52 | 40 |
| | Precuneus | LH | 581.6 | -0.19 | 0.005 | -0.27 | -5 | -57 | 31 |
| | Occipital superior cortex | RH | 544.8 | 0.17 | 0.005 | 0.27 | 29 | -63 | 36 |
| | SFG / Frontal pole | LH | 512.0 | -0.24 | 0.003 | -0.34 | -3 | 65 | 16 |
| | Orbitofrontal cortex | LH | 454.7 | -0.23 | 0.003 | -0.33 | -46 | 28 | -20 |
| | Supramarginal | RH | 438.3 | -0.12 | 0.002 | -0.18 | 54 | -33 | 32 |

analysis the dACC due to its proposed involvement in the allocation of control [30–35] as well as the right AI because of its involvement in encoding control signals and uncertainty in particular [25–28]. In order to compare the neural activity in these regions, which are generally attributed to control, with relevant processing units, we selected a region associated with mathematical calculations, the right IPS [36–38].

We explored the BOLD effect estimates ($\beta_{FIR}$) for the 2×2 balanced factorial design (satisfiability×TCC) employing Finite Impulse Response (FIR) analysis at a two-second resolution (see Section 4.9.4). We found similar patterns in AI and dACC. In both regions, the BOLD signal rose throughout the task and quickly decreased around the time the solving stage ended (Fig 3). The activity pattern in the IPS showed a different pattern to that of CON regions. In this

region, the BOLD signal increased quickly early on in the trial and was sustained until it started decreasing later on in the trial. The moment at which the decrease started was modulated by TCC and satisfiability (Fig 3).

We were also interested in studying the interaction effect between TCC and satisfiability. Conceptually, we predicted an interaction effect between complexity and proof hardness. This follows from the definition of proof hardness, which relates to the length of verifying a valid proof (proof of existence or proof of non-existence). Consequently, proof hardness is only directly informative for cases in which a participant has found (the correct) solution to verify. This entails that in instances where finding a valid witness is harder (e.g., high TCC), the effect of proof hardness should be lower on average. Specifically, we expected that instances with high TCC would have a lower differential effect on the BOLD contrast between unsatisfiable and satisfiable instances. This is exactly what we found on all three ROIs, although this effect was only consistently significant in the AI and IPS (Fig 3; fourth row of panels).

When contrasting the effect of TCC in each of the ROIs, we find that there is a significant positive effect of TCC from mid-way through the trial in the right IPS/AG (Fig 3 second row of panels). This differs from the results obtained in the whole brain analysis. Similarly, when estimating the effect of satisfiability (Fig 3; third row of panels), the results marginally differ from those of the whole-brain analysis. Firstly, the ROI analysis reveals that there is an effect of satisfiability on all three regions late in the solving-stage. Secondly, the effect of satisfiability starts in the AI and dACC mid-way through the trial. Interestingly, the effect of TCC seems to precede that of satisfiability in the IPS, whereas in the dACC the effect of satisfiability seems to precede that of TCC.

Altogether, these results suggest that both satisfiability and TCC correlate with activity in all three regions, but that their effect might have different temporal signatures. Importantly, the sign of the effect was in line with our hypothesis: a higher signal in these regions was generally related to higher TCC and unsatisfiability. The only exceptions happen briefly early on in the trial.

Finally, we explored the interaction between neural markers of accuracy and the proposed metrics of computational difficulty. We first analyzed the interaction effect between correctness and TCC on neural dynamics. We found that for instances with low TCC, there was a significant effect of correctness of the instance from early on in the trial in the IPS. Similarly, midway through the trial, a significant accuracy neural marker appeared in the AI for instances with low TCC (Fig 4; top panels). This effect was mainly due to a significantly lower BOLD signal associated with incorrect instances with low TCC. Similarly, when studying the interaction effect between correctness and satisfiability, we found a consistent significant effect of accuracy but only during satisfiable instances in the IPS, which was driven, as well, by a lower BOLD activity on incorrect instances (Fig 4; bottom panels). Importantly, this significant contrast showed from the moment the trial started.

Overall, our results suggest a link between neural markers of accuracy and metrics of computational difficulty. This relation was particularly evident in the IPS and marginally in the AI. Importantly, the differential effects of accuracy in the IPS showed from early on in the trial, suggesting that for instances associated with low computational difficulty (i.e., low proof hardness and low complexity), accuracy could be predicted from early on in the trial from BOLD activity in the IPS.

**2.2.3 Connectivity analysis.** The results so far implicate a dynamic collection of brain regions that correlate with different properties of the target computational problem at different stages of deliberation. An important open question that remains is how these regions coordinate with each other to orchestrate the intricate sequence of computations to solve the knapsack problem. In order to shed light on this coordination of computations, we studied how

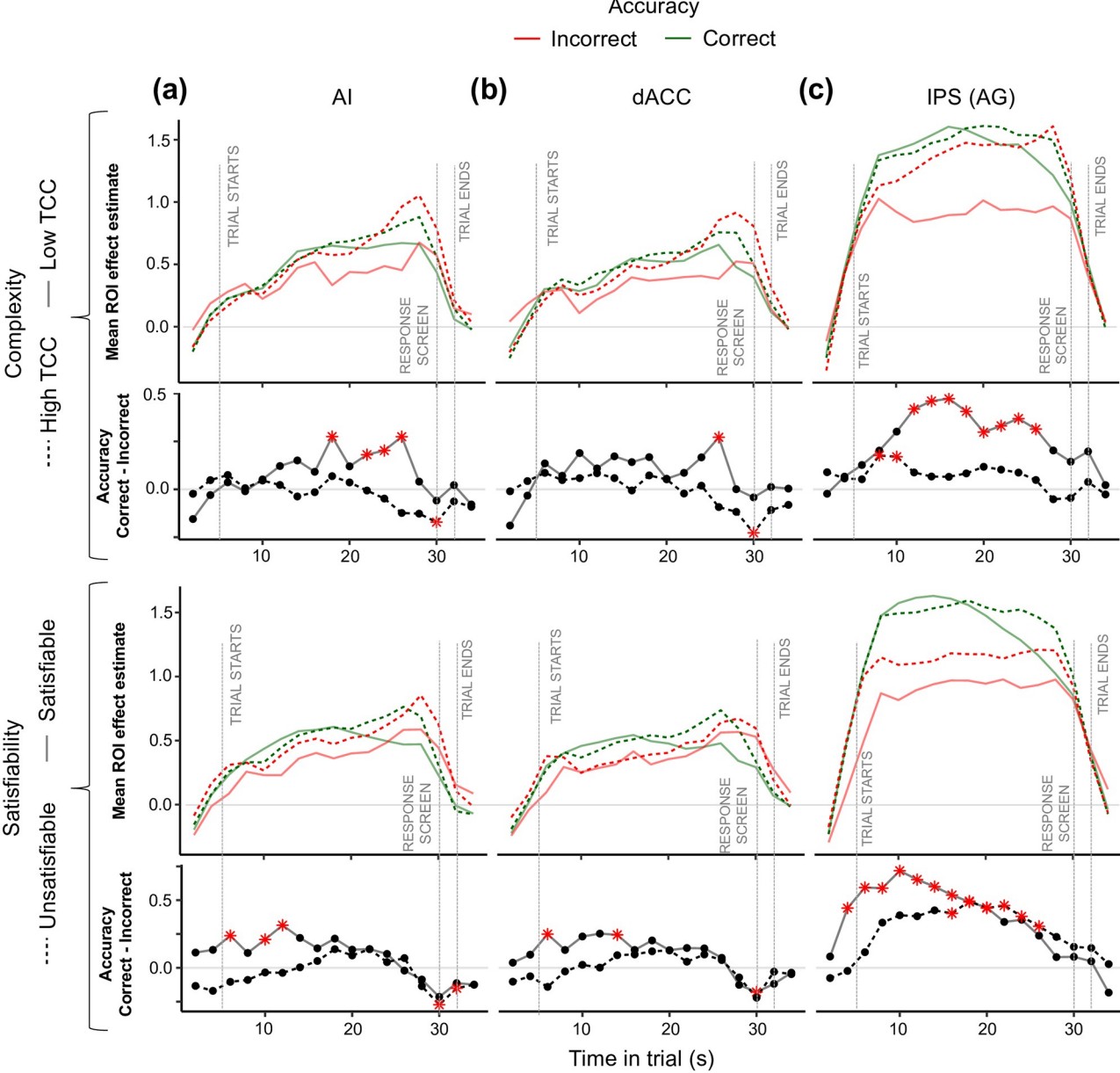

**Fig 4. Accuracy and complexity.** Mean effect estimate ($\beta$) of each ROI against time in trial. The effect at each time point represents the mean $\beta_{FIR}$ over all of the voxels from each ROI: right AI **(a)**, dACC **(b)**, and right IPS cluster extending to the angular gyrus **(c)**. The $\beta_{FIR}$'s characterize the coefficients of an FIR regression with four conditions: accuracy×TCC in the top panels, and accuracy×satisfiability in the bottom panels. The $\beta_{FIR}$ parameters are aligned to the BOLD signal, which has a lag with respect to the task time. The gray vertical lines represent the task events assuming a 5-second BOLD signal lag. Below the mean ROI effects, the second and fourth rows of figures show the accuracy contrasts ($\beta_{correct} - \beta_{incorrect}$) for different levels of TCC or satisfiability. Red stars represent significance at a 0.05 significance level.

connectivity changes throughout the problem-solving task and how these fluctuations in connectivity relate to computational difficulty (details in Section 7 in S1 Appendix).

We first studied the effect of TCC and satisfiability on *functional connectivity*. To do this, we conducted a PPI analysis to gauge the functional synchronization between each of the ROIs and other regions in the brain. Explicitly, we performed whole-brain PPI analyses employing the three considered ROIs (dACC, rAG and rAI) as seed regions. For these regressions, we

modeled the task (items and solving stages combined) with two boxcar functions of equal length (12.5s) (Fig G in S1 Appendix). This allowed us to study PPI task interactions separately for an early period (PPI-1: first 12.5 seconds of the task) and a late period (PPI-2: last 12.5 seconds). We found a similar and generalized pattern of connectivity for all three ROIs and both periods when contrasting the PPI effect compared to baseline (Fig F in S1 Appendix). This suggests that the task has a similar effect on the BOLD synchronization between the three ROIs and several other regions.

When comparing the connectivity between instances with high and low TCC, we found one significant cluster with differential connectivity. This cluster, located along the rAG and the supramarginal gyrus, showed a change in connectivity to the rAI (seed region) between high and low TCC instances during the second PPI period (Fig Ga and Table F in S1 Appendix). We also explored the differences in PPI connectivity between unsatisfiable and satisfiable instances. We observed a significant PPI effect of satisfiability between the right IPS/AG (seed) and the left MFG, as well as with the left AG, during the second PPI period (Fig Gb and Table F in S1 Appendix). Overall, these results suggest that instance properties have an effect on the synchronicity between the ROIs and a small set of other clusters. However, this effect is only significant during the later part of the solving stage. PPI analysis, however, only explores temporal connectivity. A closer inspection of the time courses in Fig 3 suggests that there may be inter-temporal relationships in activation. We turn to a study of those next.

Critical for this study, we expected the underlying neural processes of problem-solving to be internally driven. Specifically, we expected the connectivity patterns to be linked to neural processes whose timing could vary stochastically across trials and participants (e.g., the burst of neural activity does not have to coincide with an experimental intervention such as the initial display of items). In order to explore this inter-temporal connectivity (i.e., *effective connectivity*), we performed a Granger Causality (GC) analysis on the BOLD signal in these three ROIs.

We found that, throughout the task there was bi-directional effective connectivity between dACC, AI and IPS (see Fig H in S1 Appendix). Additionally, during the solving stage, we found that there was a significant change in GC from dACC to rAG. However, we did not find any significant changes in the effective connectivity between high and low TCC instances nor between unsatisfiable and satisfiable instances. Notice, however, that Granger causality only measures intertemporal correlation but not intensity. For instance, even if communication flow from, say, dACC to, say, IPS increases as a function of TCC, this will not translate into increased correlation as long as activation attributed to dACC itself increases as a function of TCC (as in Period S4 of the solving stage; see Table 1). Taken together, these results suggest that the effects of TCC and satisfiability on neural activity propagate through the ROIs via baseline effective connectivity (present during the solving stage) and not through a direct effect on the effective connectivity.

## 3 Discussion

The study of the neural underpinnings of problem-solving has, to date, been centered on tractable problems. This line of research has led to the characterization of networks and processes that support problem-solving. A critical shortcoming of existing studies is the absence of a generic theoretical framework to study the neural underpinnings of problem-solving that can be extended to intractable problems. Here, we present a framework, grounded in computational complexity theory, to study the neural underpinnings of problem-solving that overcomes previous limitations. Importantly, this theoretical framework can be applied across tasks and without knowledge of the cognitive strategies employed.

We empirically test this framework using the knapsack decision task and ultra-high field fMRI. Our findings shed light on the neural processes supporting problem-solving. Firstly, our findings not only extend but solidify the research on the neural correlates of cognitive demand by exploring the processes associated with one specific dimension of cognitive demand: computational complexity. Importantly, this is done in a task-independent way in the sense that these metrics can be applied to a whole class of problems (i.e., NP-complete). Secondly, rather than studying cognitive demand starting from "what people are thinking," we rely on the theory of computational complexity to identify intrinsic properties of a problem to delineate cognitive requirements. These intrinsic properties allowed us to discover relevant neural markers and their dynamics, similar to how risk and variance have been shown to affect decisions in probabilistic tasks [6]. Finally, the results presented provide evidence in support of the theoretical framework put forward here, which can contribute to the study of cognitive control, especially in those tasks that involve intractable problems. Critically, cognitive control involves the dynamic allocation of cognitive resources that stem from an interaction between the cognitive requirements of a task and the resources available. The framework presented here provides a theoretical foundation for the characterization of cognitive requirements that can be applied to intractable problems, which are generally understudied in the field of cognitive control.

Extensive research has studied the neural correlates of cognitive demand. This program has characterized the multiple-demand system, a network of regions that respond robustly to cognitive demand regardless of the task at hand [18–21]. This has been done using several tasks including perceptual target detection and memory retrieval, among many others. Notably, most of the tasks employed to date have been based on tractable problems. Moreover, many of the tasks modulate the cognitive demand of the task by tuning the amount of processing needed on one specific dimension of cognitive processing. For instance, in perceptual tasks, the signal-to-noise ratio is modulated [e.g., 8, 7, 39, 40]; in memory retrieval tasks, the amount of information to be stored/retrieved is adjusted [e.g., 18, 41]. The lack of a generic (problem-independent) definition of cognitive demand hinders the generalization of this approach to new problems. Here, we propose a way forward, grounded in the assumption that hardness is, at least partially, an intrinsic characteristic of the problem at hand, which can be studied across tasks employing an overarching theoretical framework.

Following this approach, we operationalized cognitive demand via TCC and found that the neural correlates of TCC overlapped with those associated with the multiple-demand system. In particular, the positively correlated clusters (higher activation in high TCC instances) in the FPN and CON resembled those of the multiple-demand system. Notably, we found clusters in the AI, the dACC, the precentral gyrus and the IPS, which have been associated with the multiple-demand system [18]. Importantly, our results display a dynamic process in which the neural correlates of TCC vary throughout the different stages of the task. This suggests that the multiple-demand system can be construed as a heterogeneous set of regions that play a dynamic and varying role at different stages in problem-solving.

It is worth highlighting that we are not arguing for the proposed framework to replace other methodological approaches in the study of cognitive demand. Instead, we assert that both approaches complement each other. Critically, complex tasks involve the interplay of several computational processing units such as working memory, logical operations, processing of numerical magnitudes among many others. Our approach, as it stands, is not able to differentiate among these sub-processes. A proper understanding of problem-solving requires both the study of these sub-processes independently, like in more classical approaches [e.g., 41], as well as in tandem in order to understand how they interact to support computationally hard problem-solving, as done in this study.

A related effect of these properties on neural processes is through the encoding of relevant task markers that could be employed during problem-solving [3, 42]. These neural markers include markers of performance such as expected error [25, 26], variance in this expectation (uncertainty) [25–27], as well as markers that encode the evidence towards a particular response [7, 43] or even the merit of alternative strategies [29, 44]. Critically, we made three conjectures with regards to these neural markers. First, we hypothesized to see markers of performance, related to TCC, from early on in the trial. Second, we conjectured we would see markers of reliability (i.e., how much can the result of a calculation be relied on to be accurate), related to satisfiability, in regions shown to encode uncertainty. Third, we expected to find neural correlates of accuracy late in the trial, which would be associated to expected performance.

With regards to our first hypothesis, we argue that TCC is a feasible metric that can be related to markers of performance and efficacy of effort from early on in the trial. Firstly, TCC has been shown to be correlated with human performance [9]. Secondly, TCC can be potentially estimated from early on in the solving stage without the need to know the solution to the problem. Indeed, TCC could be estimated by performing a straightforward summation followed by a division (add all item values and divide the result by the target profit). As such, we expected to see neural correlates of TCC from early on in the trial. Specifically, we expected to see markers of TCC from early on in the solving stage in the CON [26, 27, 30]. Contrary to our expectations, we only found significant clusters in the CON starting from the third period of the solving stage. These might reflect markers of expected performance, but other explanations cannot be excluded. For instance, this effect might reflect differences in time-on-task between TCC conditions [45]; indeed, previous work has shown that TCC affects time-on-task in other computational problems [10]. This explanation, however, would still allow these activation patterns to represent differences in neural markers such as reliability and expected performance. This follows from the fact that time-on-task is an endogenous variable of the system. That is, the agent decides when to stop reasoning about the problem, and as such, this decision would likely follow from a subjective belief on how well they can expect to perform given the current candidate solution. Therefore, differences in time-on-task between high and low TCC instances may be related to differences in subjective beliefs of both expected performance and reliability. Future work is needed to explore how and when people decide to stop deliberating, as well as the role of computational difficulty in this process.

In addition to the reported clusters that correlated positively with computational complexity, we found a set of clusters that correlated negatively with TCC. These clusters are concentrated in the second period of the solving stage, but are also found in the third and fourth periods of the solving stage. These results might be explained by the encoding of evidence accumulation signals [7, 43]. Arguably, evidence toward a solution can be accumulated faster in low TCC compared to high TCC instances. This follows directly from the theoretical prediction that the time needed to reach the solution is, on average, higher for instances with high TCC than low TCC. The difference in rate accumulation would imply that regions that encode evidence accumulation would show a higher activation on low TCC instances early in the trial, in accordance with the pattern found in the second period of the solving stage.

Turning now to our second conjecture regarding the neural markers of reliability, we explored the correlates of satisfiability during problem-solving. We expected to see activation related to satisfiability in regions previously associated with uncertainty encoding, specifically in the CON. In line with our hypothesis, we found a significant positive relation between unsatisfiability and activity in the CON that started halfway through the solving stage. In line with our conjecture, we found that the neural markers of satisfiability overlap with regions that encode probabilistic uncertainty. This suggests that reliability and uncertainty might constitute

analogous constructs that are encoded similarly across tasks and that could serve a generic role in decision-making. While our findings provide a potential explanation for the observed neural activity, further work would be needed to categorically connect these constructs across tasks. The CON is associated with a multitude of cognitive processes, and the activity we observed could be linked to any number of these. Therefore, our interpretation, while supported by our data, should not be seen as definitive. Further research is necessary to connect these constructs across various tasks definitively and to explore other potential explanations for the observed neural activity.

Contrary to our expectations, we found several regions that displayed an increase in activity during satisfiable instances from early on in the solving stage. This result is perplexing because knowing the satisfiability of the problem equates to having solved the problem, which would not be expected early on in the trial. A possible explanation for this is that the clusters found encode evidence accumulation [7, 43] and that accumulating evidence towards the solution in satisfiable instances occurs at a different rate than in unsatisfiable instances. However, this is not directly supported by theory. The computational difficulty of finding a solution is determined by TCC, and the selection of instances in this study ensured that TCC remained balanced between satisfiable and unsatisfiable instances. This entails that, on average, the expected level of complexity (and evidence accumulation rate) would be the same for both satisfiable and unsatisfiable instances. Alternatively, these activation patterns might reflect the use of different strategies. For example, if an individual predicts that a given instance could potentially be satisfiable, they could search for a subset of items that meets the defined constraints. However, if the perception is that the given instance might be unsatisfiable, then the individual may start a search for proof of infeasibility, like verifying (in an extreme case) whether fitting even a single item into the knapsack is impossible due to all the item weights surpassing the allowed weight capacity. However, this account would still require participants to implement different strategies, based on satisfiability, as early as during the first few seconds of the solving stage. Overall, future research should attempt to disentangle the effect of proof hardness from that of satisfiability and complexity. This could be done, for instance, through experiments aimed at testing more nuanced metrics of proof hardness beyond worst-case complexity classes.

Moving on now to consider our conjecture related to the neural markers associated with erring, we explored the effect of accuracy on neural activation throughout the task. It has been proposed that FPN and CON regions encode task signals related to error detection and error expectation [25, 27, 31]. We hypothesized that participants would represent a subjective belief on the expected accuracy (or reward) of their answer (e,g, [29]). In line with our conjecture, we found that activity in both the FPN and CON was positively correlated with erring during the response stage, even in the absence of feedback, as was the case in our design (Section 6 in S1 Appendix).

A puzzling finding in this regard is the fact that neural correlates of accuracy are identified early on in the trial in the ROIs, especially in instances with low computational difficulty (i.e., low proof hardness and low complexity). One possible explanation for this is attentional engagement on the task. If a participant does not actively engage in the task, they are more likely to settle on an incorrect solution. In turn, the likelihood of reaching an incorrect answer due to inattention is higher among instances with low computational difficulty. Together, these patterns would partially explain the marked difference in BOLD activity between correct and incorrect trials in the IPS. However, other alternative explanations are possible. Further work is needed to fully identify the dynamics of effort and attention allocation in computationally complex tasks.

Overall, we found evidence that suggests the existence of neural markers related to computational complexity, proof hardness and performance. Taken together, the framework put forward here provides a way to study neural markers associated to subjective beliefs during problem-solving. It is worth noting, however, that while we modulated complexity and proof hardness, many other complexity-related features might be relevant, including, for example, the size of the problem at hand [e.g., 46–51]. Further work in this area is needed to understand the interaction between different sources of computational difficulty in human problem-solving.

Finally, to explore the dynamics related to control during complex problem-solving, we analyzed the functional interaction during problem-solving of three ROIs, two of which have been associated with cognitive control (i.e., CON) and one region which has been associated with processes that were deemed highly relevant for the task at hand (i.e., IPS). We studied synchronization of signals (employing PPI analysis) and explored their effective connectivity (using GC analysis).

Our results support the view that there is a generalized change in signal synchronization during the solving stage compared to baseline. Moreover, when exploring the link between instance properties and synchronicity between regions, we found several clusters whose connectivity was modulated by either satisfiability or TCC. These effects were only present late in the trial. Specifically, we found that TCC modulated the synchronicity between the rAI and the rIPS. Additionally, satisfiability modulated the functional connectivity between the right IPS and two clusters in the left hemisphere, one in the AG and one in the MFG. Overall, these results suggest a differential recruitment of regions during the task, partially modulated by task properties late in the trial. Interestingly, the significant clusters identified in this analysis have been implicated in the performance of mathematical calculations [38, 52], suggesting that they could support moment-to-moment implementation of strategies. Further work would be needed in order to assess whether the relation, found here, between instance properties and functional synchronization is associated to the implementation of different strategies.

Additionally, we found that the effective connectivity pattern was impervious to the level of TCC and satisfiability. This suggests that the impact of computational complexity on control would operate by inducing varying levels of activity within specific regions of interest, rather than through the modulation of the effective connectivity between these regions. In other words, our findings reveal that while the effective connectivity, which signifies the correlation in activation among subregions, remains stable, there is a distinct alteration in the flow of information. This alteration arises from changes in activation within the nodes (i.e., regions) while the underlying connectivity between them remains unaltered.

\*\*\*

Humans are constantly solving problems that vary in complexity, ranging from perceptual tasks, such as motion detection and face recognition, to reasoning tasks, such as choosing an investment portfolio. Understanding how the complexity of these problems affects the neural processes involved in problem-solving is of crucial importance for the understanding of human decision-making. Here, we present a framework that allows for the study of the computational difficulty of human problem-solving. We applied this framework and identified a dynamic set of regions in which activation was modulated by different properties related to computational complexity. Overall, our findings provide support to the premise that computational complexity theory, as applied here, provides a useful characterization of cognitive demand and reliability for the study of problem-solving in neuroscience.

## 4 Materials and methods

### 4.1 Ethics statement

The experimental protocol was approved by the University of Melbourne Human Research Ethics Committee (Ethics ID 1749616.3). Written informed consent was obtained from all participants prior to the commencement of the experimental sessions. Experiments were performed in accordance with all relevant guidelines and regulations.

### 4.2 Participants

Twenty right-handed volunteers from Melbourne University and the surrounding community took part in the study (14 female, 5 male, 1 other; age range = 18–35 years, mean age = 26.6 years). Inclusion was based on age (minimum = 18 years, maximum = 40 years) and on right-handedness. Each participant performed the knapsack decision task in the scanner and performed outside the scanner the knapsack optimization task and a set of basic cognitive function tasks.

### 4.3 Knapsack decision task

In this task, participants were asked to solve a number of instances of the (0–1) knapsack decision problem (Fig 1a). In each trial, they were shown a set of items with different values and weights, as well as a capacity constraint and a target profit. Participants had to decide whether there exists a subset of those items for which (1) the sum of weights is lower or equal to the capacity constraint and (2) the sum of values yields at least the target profit.

Each trial had four stages. In the first stage (items stage; 3 seconds), only the items were presented. Item values, in dollars, were displayed using dollar bills and weights, in grams, were shown inside a black weight symbol. The larger the value of an item, the larger the dollar bill was in size. Similarly, the larger the weight of an item, the larger its weight symbol was in size. At the center of the screen, a green circle indicated the time remaining in this stage. In the second stage (solving stage; 22 seconds), target profit and capacity constraint were added to the screen inside the green timer circle. In the third stage (response stage; 2 seconds), participants saw a 'YES' and a 'NO' button on the screen, in addition to the timer circle, and made a response using the keyboard (Fig 1a). Finally, a jittered inter-trial rest period of 8, 10 or 12 seconds was shown before the start of the next trial.

Participants completed 56 trials (7 blocks of 8 trials), each showing a different instance of the knapsack decision problem. The order of instances was randomized across participants. The side of the 'YES' and 'NO' buttons was also randomized.

### 4.4 Complexity, proof hardness and instance sampling

We modulate the complexity of solving an instance using typical-case complexity (TCC). In the knapsack problem, TCC is explicitly connected to a set of parameters $\bar{\alpha} = (\alpha_c, \alpha_p)$ that capture the constrainedness of the problem [9, 14]. For a knapsack instance with $N$ items with weights $w_i$, values $v_i$, capacity constraint $c$ and target profit $p$, the constrainedness parameters are defined as:

$$\alpha_p = \frac{p}{\sum_{i=1}^{N} v_i} \tag{1}$$

$$\alpha_c = \frac{c}{\sum_{i=1}^{N} w_i} \tag{2}$$

These parameters determine the likelihood that a random instance is *satisfiable*, that is, that the solution is 'yes'. Specifically, they characterize where typical instances are generally satisfiable (under-constrained region), where they are unsatisfiable (over-constrained region) and where the probability of satisfiability is close to 50% (satisfiability threshold $\alpha_s$). It has previously been shown that the computational difficulty of solving the problem is higher when $\bar{\alpha}$ is close to $\alpha_s$ [9, 14].

Fixing $\alpha_c$ allows us to modulate the complexity by varying the levels of $\alpha_p$. TCC is explicitly defined based on the distance of $\alpha_p$ to the satisfiability threshold $\alpha_s$. Instances with values of $\alpha_p$ near the satisfiability threshold have a high typical-case complexity (*high TCC*) whereas instances further away from it—that is, in the under-constrained and over-constrained regions —have low typical-case complexity (*low TCC*). It is worth noting that the generalizability of TCC relies on the covert assumption that the function of TCC in relation to $\bar{\alpha}$ is not influenced by the sampling method (which seems like a reasonable assumption under the condition that sampling itself is not dependent on $\bar{\alpha}$).

We modulate proof hardness using insights from canonical computational complexity classes. For NP-complete problems (like the knapsack decision problem), this theory predicts that it is easier to prove that a solution is correct if the instance is satisfiable than if it is unsatisfiable. For example, in the case of the knapsack, for a *satisfiable* instance (the correct choice is 'yes'), it suffices to find a subset of items that satisfies the weight capacity and value constraints. Subsequently, verification that this subset of items satisfies the constraints can be done in polynomial time. In contrast, to confirm that an instance is *unsatisfiable* (the correct choice is 'no') requires proving that no subset of items exists that satisfies the constraints. It is conjectured and broadly accepted that verifying such a proof is not in P (the polynomial time complexity class). Theoretically, the asymmetry reflects the conjectured null intersection between complexity classes NP-complete and co-NP-Complete (Fig 5).

Instances were sampled following a 2×2 balanced factorial design (Fig 6) for the factors TCC (high and low) and satisfiability (satisfiable and unsatisfiable). Specifically, instances were sub-sampled from those employed in a previous behavioral study [9]. Instances in that study were selected such that $\alpha_c$ was fixed ($\alpha_c \in [0.40, 0.45]$) and the instance constrainedness varied according to $\alpha_p$. 18 satisfiable instances were selected from the under-constrained region ($\alpha_p \in [0.35, 0.4]$; *low TCC*) and 18 unsatisfiable instances from the over-constrained region ($\alpha_p \in [0.85, 0.9]$; *low TCC*). Note that, by definition, sampling from the underconstrained region is unlikely to generate unsatisfiable instances, and analogously, sampling from the overconstrained region is unlikely to generate satisfiable instances. We leveraged this fact to obtain a balanced design in which half of the instances were satisfiable and half unsatisfiable. Additionally, 18 satisfiable instances and 18 unsatisfiable instances were sampled near the satisfiability threshold ($\alpha_p \in [0.6, 0.65]$; *high TCC*). Half of the instances with high TCC were forced to have high/low computational requirements (top/bottom 50%), according to an algorithm-specific ex-post complexity measure of a widely-used algorithm (Gecode [53]). All instances in the experiment had $N = 6$ items and $w_i$, $v_i$, $c$ and $p$ were integers.

In the current study, we randomly selected 56 of the 72 instances sampled in Franco et al. [9]. Sub-sampling without replacement was done ensuring that the same number of instances were selected across TCC and satisfiability conditions. Moreover, instances with high TCC were balanced to require high/low computational requirements according to the same algorithm-specific complexity measure employed in their study (i.e., Gecode propagations).

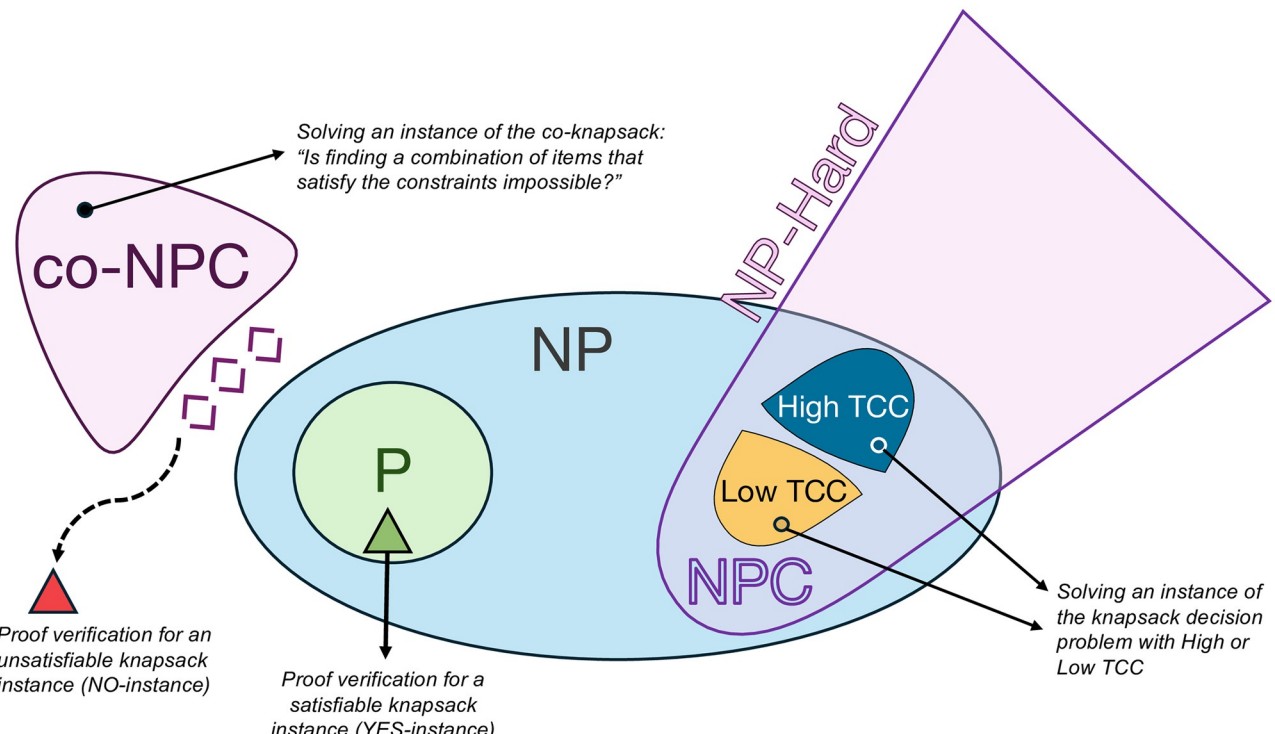

**Fig 5. Complexity classes.** The knapsack decision problem belongs to the class NP-Complete (NPC) because it satisfies the dual-qualifying criteria of NP and NP-hard. It is NP, given that it fulfills the NP defining condition: a YES-certificate of a satisfiable instance can be verified in polynomial time (P). It is NP-hard since it is at least as hard as any other problem in NP. It is conjectured that P≠NP, which entails that the NPC problems are not solvable in polynomial time (i.e., they are harder and require more computational resources—time—to solve than problems in P). Within the class of NPC problems, there are instances that are harder than others. A key discriminator factoring instances by the respective computational resources needed for their resolution is their typical-case complexity (TCC). The class noted as co-NP-Complete (co-NPC) comprehends problems such as the co-knapsack. The aim of this problem is to determine if the existence of a subset of items that satisfy the constraints is infeasible. Every satisfiable knapsack instance has a counterpart unsatisfiable co-knapsack instance. It is conjectured that co-NPC is not in NP, thereby implying that verifying a proof of non-existence for an unsatisfiable knapsack instance is not in P; it is harder.

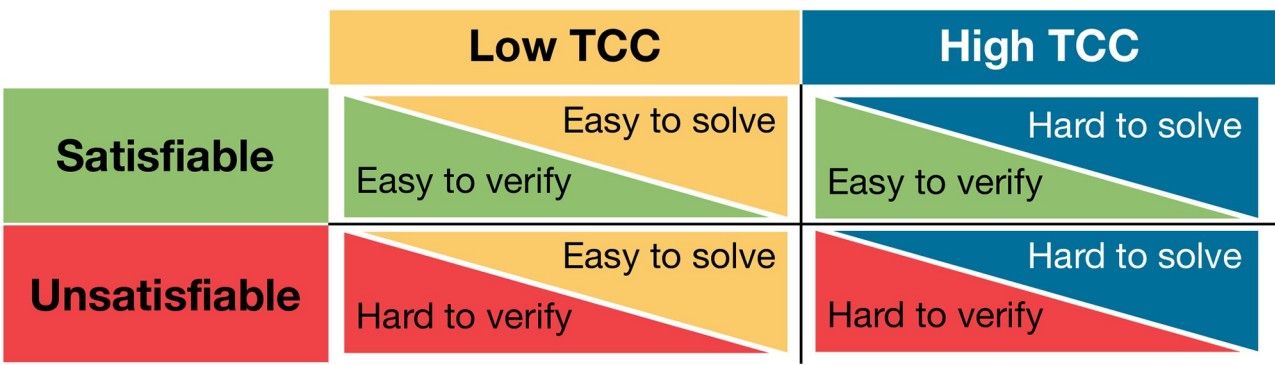

**Fig 6. Instances sampled.** Instances were sampled using a 2x2 factorial design, ensuring that each participant answered an equal number of instances for each of the four possible categories of TCC and proof hardness. Each category presents a distinct profile of proof hardness (the computational difficulty of validating a solution) and complexity (the computational difficulty of solving the instance).

## 4.5 Complementary tasks

Participants were presented with a set of complementary tasks outside of the scanner. They were asked to solve a number of instances of the (0–1) knapsack optimization problem. Similar to the knapsack decision task, participants were shown a set of items with different weights and values as well as a capacity constraint. However, unlike the decision variant, no target profit was presented. Participants had to find the subset of items that *maximized* total value subject to the capacity constraint (see Section 3 in S1 Appendix).

We also tested five cognitive capacities that we considered relevant for the knapsack tasks, namely, working memory, episodic memory, strategy use, processing and psychomotor speed, as well as mental arithmetic. To do so, we administered a set of tasks from the Cambridge Neuropsychological Test Automated Battery (CANTAB; see Section 5 in S1 Appendix). Specifically, we asked participants to perform the Reaction Time (RTI), Paired Associates Learning (PAL), Spatial Working Memory (SWM) and Spatial Span (SSP). In addition, participants were presented with a set of mental arithmetic problems (Section 5 in S1 Appendix).

## 4.6 Procedure

Participants were asked to fill in an MRI screening form before attending the experiment. Once at the experiment, participants were presented with a plain language statement and a consent form. After reading these and providing written informed consent, participants were instructed in the tasks and completed a practice session of the knapsack decision task. Participants then underwent an MRI safety check.

Before being scanned, participants solved the CANTAB RTI task outside of the scanner. This was followed by the scan session in which they performed the knapsack decision task. Afterwards, outside of the scanner, they completed the CANTAB RTI task again, followed by the knapsack optimization task. Subsequently, they completed the remaining CANTAB tasks in the following order: PAL, SWM and SSP. Finally, they performed the mental arithmetic task and completed a set of demographic and debriefing questionnaires. Altogether, the experimental session lasted around three hours.

Participants received a show-up fee of A$10, as well as monetary compensation based on performance. They earned A$1.2 for each correct answer in the knapsack decision task and for each correct answer in the knapsack optimization task.

## 4.7 Behavioral statistical analyses

The R programming language was used to analyze the behavioral data. All of the linear mixed models (LMM), generalized logistic mixed models (GLMM) and censored linear mixed models (CLMM) included random effects on the intercept for participants (unless otherwise stated). Different models were selected according to the data structure. GLMM were used for models with binary dependent variables, LMM were used for continuous dependent variables and CLMM were used for censored continuous dependent variables (e.g., time-on-task).

All of the models were fitted using a Bayesian framework implemented using the probabilistic programming language Stan via the R package 'brms' [54]. Default priors were used. All population-level effects of interest had uninformative priors; i.e., an improper flat prior over the reals. Intercepts had a student-t prior with 3 degrees of freedom and a scale parameter that depended on the standard deviation of the dependent variable after applying the link function. The t-student distribution was centered around the mean of the dependent variable. Sigma values, in the case of Gaussian-link models, had a half student-t prior (restricted to positive values) with 3 degrees of freedom and a scale parameter that depended on the standard deviation

of the dependent variable after applying the link function. Standard deviations of the participant-level intercept had a half student-t prior that was scaled in the same way as the sigma priors.

Each of the models presented was estimated using four Markov chains. The number of iterations per chain was, by default, set to 2000. This parameter was adjusted to 4000 for some models to ensure convergence, which was verified using the convergence diagnostic $\hat{R}$. All models presented reach an $\hat{R} \approx 1$.

Statistical tests were performed based on the 95% credible interval estimated using the highest density interval (HDI) of the posterior distributions calculated via the R package 'parameters' [55]. For each statistical test we report both the median ($\beta_{0.5}$) of the posterior distribution and its corresponding credible interval ($HDI_{0.95}$).

No participant nor trial was excluded from the data analysis of the knapsack decision task.

## 4.8 MRI data acquisition

We collected the fMRI images using a 7 Tesla Siemens MAGNETOM scanner located at the Melbourne Brain Centre (Parkville, Victoria) with a 32-channel radio frequency coil.

The BOLD signal was measured using a multiband echo-planar imaging sequence (TR = 800 ms, TE = 22.2 ms, FA = 45˚). We acquired 84 interleaved slices (thickness = 1.6 mm, gap = 0 mm, FOV = 208 mm, matrix = 130x130, multi-band factor = 6, voxel size = $1.6 \times 1.6 \times 1.6 mm^3$) per volume. 380 volumes were acquired on each run while recording cardiac and respiratory traces.

After five functional runs (one resting state run followed by four task runs), a high resolution (0.7 mm isotropic) anatomical image was acquired using an MP2RAGE pulse sequence (TR = 5000 ms, TE = 3.07 ms, TI1 = 700ms, FA1 = 4˚, TI2 = 2700ms, FA1 = 5˚, matrix = 330×330, voxel size = 0.73×0.73×0.73mm$^3$, FOV = 240 mm, 224 slices, slice thickness = 0.73). Afterwards, another three functional runs were performed, followed by a diffusion weighted imaging (DWI) multiband sequence (TR = 7000 ms, TE = 72.4 ms, FA = 90˚, FoV = 210 mm, matrix = 170x170, slice thickness = 1.24, voxel size = $1.24m^3$, 128 slices, multiband factor = 2).

## 4.9 Imaging statistical analyses

**4.9.1 Preprocessing.** Initial preprocessing of the data was performed using AFNI [56] and the Advanced Normalization Tools (ANTs) software. For each subject, pulse and cardiac noise was regressed out from the functional scans. These were then slice-time corrected and the volumes were motion-corrected by registering them to the first volume of the first functional run. The mean image of the first run was co-registered to the anatomical scan (down-sampled) and this transformation was applied to all of the functional volumes. Afterwards, each participant's anatomical scan was used for calculation of transformation parameters to normalize the functional images into the Montreal Neurological Institute (MNI) space (see Section 1 in S1 Appendix for more details).

**4.9.2 Whole-brain analysis (boxcar).** Whole-brain analyses were performed by fitting generalized linear models (GLM) using AFNI [56]. Before the regressions were implemented, we spatially smoothed the functional volumes with a 4.8mm FWHM Gaussian kernel. Additionally, volumes with motion or signal outliers were censored from each of the regressions.

We performed GLM regressions to explore three contrasts of interest. Specifically, we tested the neural correlates of TCC (high TCC vs. low TCC), satisfiability (unsatisfiable vs. satisfiable) and accuracy (correct vs. incorrect). In each of the regressions, the solving phase (22s) was

modeled using four boxcar functions of equal duration (5.5s):

$$y = \quad \beta_0 + \sum_{i=1}^{4} [\beta_i^{L0} L_0 \times box_{Si} + \beta_i^{L1} L_1 \times box_{Si}] + \beta_5^{L0} L_0 \times box_{resp} + \beta_5^{L1} L_1 \times box_{resp} + \\ \beta_6 box_{items} + \beta_L Left + \beta_R Right \tag{3}$$

where $L_0$ and $L1$ correspond to the different levels of interest (e.g., high TCC and low TCC respectively) and $box_{Si}$, $box_{resp}$ and $box_{items}$ correspond to the boxcar functions of the solving, response and items stages, respectively. *Left* and *Right* correspond to the button pressed by the participant.

Group level analyses were performed using mixed effects multilevel modeling [57]. All whole-brain analysis results are reported with a clusterwise threshold of $p < 0.05$ corrected for multiple comparisons across the whole brain, using an uncorrected voxelwise threshold of $p < 0.001$.

**4.9.3 ROI specification.** We were particularly interested in how control and subjective beliefs of cognitive demand and reliability were involved in complex problem-solving. To study these dynamic processes we selected three regions of interest (ROIs) that have been implicated in the processes of interest. Firstly, we included in our analysis the CON (dACC and AI) due to its proposed involvement in the allocation of control [30–35] and uncertainty encoding [25–27], which we conjectured would be highly related to encoding of reliability. Secondly, we included a region that has been involved in moment-to-moment processing operations during problem-solving. We expected the knapsack task to engage processing units associated with number processing and mathematical calculations. Therefore, we selected a region that has been widely connected to 'processing' in mathematical problem-solving, the right IPS [36–38].

The three ROIs were selected from the clusters found when contrasting high and low TCC in the last boxcar during the solving stage (period S4). We chose the contrast for the fourth boxcar for a few reasons. We expected that during this last period of the solving stage we would be able to see a marked differentiation in the cognitive demand between instances with high and low TCC. We expected instances with low TCC to require less computational time and thus, we hypothesized that, on average, participants would still be making calculations during the period S4 for high TCC instance, but not for low TCC instances. This was further indicated by a parallel pilot study that found that participants spent on average 17.9s solving an instance with low TCC and 21.2s on those with high TCC (period S3 ends at 19.5s of solving stage). Importantly, we believed that these differences in cognitive demand would be reflected as well in a differentiation in the control activity in the system. Critically, we expected the monitoring of control variables such as expected performance would differ between types of instances. For instance, we expected the subjective markers of performance would converge to actual performance levels in the late stages of the solving stage ([9]; Fig 1), which would imply higher subjective beliefs of expected performance for low TCC. Additionally, we expected that this contrast would allow us to control for task-set signals [23]. We conjectured that the task-set signals would be maintained during the whole solution stage, so the proposed contrast would not capture task-set signals encoding goals nor the underlying structure of the task.

Among the significant clusters found around the right IPS, we chose the IPS (AG) cluster (peak: x = 32, y = -65, z = 47) because of its overlap with the regions that were found to be associated with mathematical calculations in a meta-analysis [38].

**4.9.4 ROI temporal dynamics.** We explored the dynamics in these ROIs by fitting generalized linear models (GLM) using AFNI [56]. Analogous to the whole brain GLM analysis (i.e., boxcar analysis), we spatially smoothed the signal and censored outliers from the regression.

In this case, in contrast to the whole brain analysis GLMs, we modeled the trial time using a Finite Impulse Response (FIR) approach, in which each trial was modeled using 17 simple basis functions (tents).

This approach allowed us to take advantage of the short TRs (0.8s) used for the functional acquisition sequence, which were possible due to the ultra-high-field MRI used in the experiment. Modeling the BOLD signal using FIR allowed us to obtain 17 beta estimates $\beta_{FIR}$ for each voxel for each of the conditions considered. Note that these estimates model the hemodynamic response directly and, therefore, they do not factor in the lag of the BOLD signal. In order to link each $\beta_{FIR}$ to a time in the task, we assumed a lag of 5 seconds in the hemodynamic response.

We obtained a 2×2 $\beta_{FIR}$-estimates for the factors TCC (high and low) and satisfiability (satisfiable and unsatisfiable). We explored the dynamics of each ROI by estimating the average $\beta_{FIR}$ over all of the voxels from each ROI for each condition. The ROI signal aggregation was performed using python 3.7 and the nilearn library.

## Supporting information

**S1 Appendix. Supplementary methods and results.**
(PDF)

## Acknowledgments

The authors would like to acknowledge Rebecca Glarin and Scott Kolbe for their assistance in the planning and successful execution of the MRI scans.

## Author Contributions

**Conceptualization:** Juan Pablo Franco, Peter Bossaerts, Carsten Murawski.

**Data curation:** Juan Pablo Franco.

**Formal analysis:** Juan Pablo Franco, Peter Bossaerts, Carsten Murawski.

**Funding acquisition:** Carsten Murawski.

**Investigation:** Juan Pablo Franco, Peter Bossaerts, Carsten Murawski.

**Methodology:** Juan Pablo Franco, Peter Bossaerts, Carsten Murawski.

**Project administration:** Juan Pablo Franco, Carsten Murawski.

**Resources:** Carsten Murawski.

**Software:** Juan Pablo Franco, Peter Bossaerts, Carsten Murawski.

**Supervision:** Peter Bossaerts, Carsten Murawski.

**Validation:** Peter Bossaerts, Carsten Murawski.

**Visualization:** Juan Pablo Franco.

**Writing – original draft:** Juan Pablo Franco, Peter Bossaerts, Carsten Murawski.

**Writing – review & editing:** Juan Pablo Franco, Peter Bossaerts, Carsten Murawski.

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
