## [Decision Letter · Decision Letter 0]

16 Apr 2024

Dear Dr. Murawski,

Thank you very much for submitting your manuscript "The neural dynamics associated with computational complexity" for consideration at PLOS Computational Biology.

As with all papers reviewed by the journal, your manuscript was reviewed by members of the editorial board and by several independent reviewers. In light of the reviews (below this email), we would like to invite the resubmission of a significantly-revised version that takes into account the reviewers' comments.

We cannot make any decision about publication until we have seen the revised manuscript and your response to the reviewers' comments. Your revised manuscript is also likely to be sent to reviewers for further evaluation.

Sincerely,

Daniele Marinazzo

Section Editor

PLOS Computational Biology

Reviewer's Responses to Questions

**Comments to the Authors:**

Reviewer #1: In this study, the author study the neural correlates of computational hardness. They use the knapsack problem in which a solution is searched to satisfy a set of constraint. Computational difficulty was manipulated by comparing problems that are satisfiable and unsatisfiable (the latter being harder to demonstrated because it is necessary to demonstrate that no solution exist, whereas finding only one solution demonstrates satisfiability). They also manipulated difficulty by manipulating the likelihood of a random instance of the problem satisfying the constraint, which they term “typical case complexity”. They find that (human) subjects’ performance decreases with typical case complexity, but does not change depending on satisfiability. At the neural level, they find that typical case complexity and unsatisfiability correlate with increased activity in a number of regions, including the dorsal anterior cingulate cortex, anterior insula and intra parietal sulcus. They show that these effects typically build up over time within a given problem solving period. They report that these two factors also modulate connectivity between these regions.

This study makes a useful contribution in uncovering some aspects of reasoning in complex task. It benefits from a computational characterization of the task and interesting behavioral results (but see also my concerns below). It is clearly written, thoroughfully analyzed (see the impressive number of appendices), it uses a variety of methods (neural correlates, analysis of temporal dynamics, change in temporal connectivity and directed connectivity), and the methods are generally solid. I list a number of concerns that could improve the paper if addressed in a revision.

Major concerns:

1) Behavioral relevance of satisfiability

The main section reports that there is no behavioral effect of satisfiability, but some neural effects of satisfiability. It would be more convincing if the authors could show some behavioral effect of satisfiability, which would indicate that this concept accounts for the way human subjects solve the task. If I understood correctly, it should take longer to answer the problem when it is not satisfiable, because when it is, it suffices to find one solution to conclude that the problem is satisfiable. In some self-pace version of the task, a difference could be found in reaction times. I apologize if this effect is reported in appendix (I looked for it…) or in a previous publication; in either case it would be useful to draw the attention of the reader to it.

2) Just about brain mapping?

The study is framed mostly as a brain mapping study: what regions correlate with typical case complexity and satisfiability? There is a bit of connectivity (PPI and Granger) but the results are not extremely informative. I would encourage the authors to motivate more their approach and the conclusions that we can draws from it. The main conclusions of the paper seem to be related to several forms of reverse inference. However, it is often the case that many concepts or task factors or effects are associated with activity in a given brain region. This is particularly a concern for variables whose neural correlate is expected to not be very specific, e.g. when looking for regions (l. 220) “linked to monitoring of uncertainty” (which is expected to also typically reflect other things such as effort, difficulty, attention). To be clear, the comment is not about new analyses, but improving a bit the framing of the paper; it could improve it significantly.

3) Put more behavioral results in the main text

I would recommend putting the appendix C.1 in the main text for two reasons. First, it introduces the measure of “instance complexity” which is advantageously defined at the instance level, whereas TCC is defined for a group of (randomly selected) instances. Second, it offers a richer description of the data, in particular, Fig 7 shows the increase in performance with instance complexity (as the name does not suggest, instance complexity is smaller when the problem is more difficult to solve), and the absence of effect of satisfiability. I understand that this figure replicates the results of a previous paper, but it gives more confidence about the behavioral effects in this specific study.

4) fMRI analysis

The task design and analysis are essentially categorical. In the case of complexity, would not it be more powerful to do a regression analysis of fMRI signals with the instance complexity (defined for each instance, appendix C) rather than with high/low TCC (which, in addition to being binarized, is defined only at the level of a group of randomly sampled instance)?

Minor:

- l. 138: A formalism is introduced to define constrainedness and satisfiability, but the formalism is not very useful because it is not explained. What is p, c, v_i, w_i, N? I think I guessed but it would be better to have them explained. I would recommend to either remove the corresponding section of the results (that explains the method) and simply claim that you have a formalism for constrainedness and satisfiability, or to explain it in a way that is self contained (instead of referring the reader to you previous paper, Franco 2021). If such a section is long, then it could appear in the Method section. Currently, the Method section does not explain how constrainedness and satisfiability are computed. Section 4.4 of the Method is also impossible to understand without having been introduced with the formalism of alpha_p and alpha_c.

- The paper repeatedly reports an absence of progression during the task (e.g. no change in performance as the number of trials increases). However, the confidence interval is close to detecting an effect, with 0 (no effect) being very close to the lower bound of the interval. Could it be that the change over trials is not linear but gradually saturates, and would be better captured by a regression with the square-root of the number of trials?

- l. 189: “We found that the neural correlates of TCC varied throughout the duration of the solving stage (Fig 2a, Table 1).” It is difficult to see some dynamics in this figure because maps are thresholded, therefore some small changes in the effect can have dramatic effects in the figure (e.g. when crossing the threshold for significance). I would recommend using unthresholded maps for this figure (together with Table 1 which reports significance levels corrected for multiple comparisons).

Reviewer #2: This study uses a formal definition of computational complexity to examine the neural mechanisms of complexity processing in humans. While being scanned in a 7T MRI, participants engaged in a decision making task that dissociates problem complexity from choice difficulty. The results implicate a network of interacting brain areas including dorsal anterior cingulate cortex, the anterior insula and the angular gyrus.

The study’s major strength is its formal approach to studying complexity. The fMRI analyses are appropriate and relatively straightforward. Overall I believe that this paper will make an important contribution to the literature. That said, in my view the main ideas are described rather abstrusely, especially for a non-expert like me. The paper quickly serves up a word salad of terms such as NP-complete, NP-hard, NP, complexity, proof hardness, random ensembles of instances, typical-case complexity, satisfiability, co-NP-complete, reliability, and constrainedness. I imagine that these concepts may be familiar to many computer scientists, but for an interdisciplinary journal more could be done to introduce this terminology. My comments below reflect this opinion.

1. Complexity is measured using “random ensembles of instances”. This technique seems to have been applied mainly by the authors themselves in a few of their recent studies, in order to derive what they call “typical-case complexity”. Given that this technique is not yet widely used in neuroscience, it should be described here in greater depth.

2. Brief definitions of NP-complete and co-NP-complete are warranted, as well as descriptions of how these terms relate to NP-hard and NP problems.

3. Constrainedness is defined according to equations for alpha-p and alpha-c (line 138), but the parameters in these equations are undefined: what are p, v, I, c and w?

4. Figure 1b indicates that this task is characterized by essentially four types of computational problems: 1) low TCC and satisfiable (“underconstrained”, low complexity); 2) low TCC and unsatisfiable (“overconstrained”, high complexity), 3) High TCC and satisfiable (medium complexity); and 4) high TCC and unsatisfiable (medium complexity). It would be helpful if an example trial were provided for each of these four conditions. I am particularly interested in the high complexity trials, which are both low typical-case complexity and unsatisfiable. What is it about these trials that make them overconstrained and therefore easy to solve, despite being unsatisfiable? Further, why is it not possible to have low complexity trials that are unsatisfiable and high complexity trials that are satisfiable? An example for each of the 4 conditions would go a long way toward making the entire paper more transparent.

5. Regarding the fMRI results, can the authors compare the underconstrained vs. overconstrained conditions? Are these results different from the satisfiable vs. unsatisfiable contrast, given that the high TCC trials are removed from the comparison?

6. Can the authors speculate more about the cognitive processes involved in the task? It is a bit difficult to understand what is going on without some kind of theoretical model to anchor the data on. To be sure, these issues are touched on in the paper — for example, it states that “TCC can be potentially estimated from early on in the solving stage without the need to know the solution to the problem” (line 406), and “evidence toward a solution can be accumulated faster in low TCC compared to high TCC instances (line 425) (see also lines 437-448) — but in my view these ideas could be fleshed out even more. How can people estimate TCC? And depending on TCC level, what strategies do they apply, and does it depend on the level of complexity?

For example, presumably at the start of each trial subjects use some kind of general purpose strategy, whereas later in the trial they tailor or switch strategies according to the specific problem type. Most interesting is that although performance is relatively good on all low TCC trials, the hemodynamic response suddenly decreases for the satisfiable (underconstrained) trials but not the unsatisfiable (overconstrained) trials. Am I right to assume that although accuracy is about the same for both underconstrained and overconstrained trials, RTs are much slower for the overconstrained relative to the underconstrained trials? These are issues that a processing model could help illuminate. (I suggest that Figure 1B also show the RTs in addition to accuracy).

7. In my view the results of the PPI/Granger causality analyses are not very informative. So far as I can see, the analyses show that three somewhat arbitrary ROIs interact with each other as a function of task complexity. Mightn’t we have guessed this going into the experiment? What have we learned? Whether or not these analyses should remain in the paper is a judgement call, but I’d advocate for removing them.

**Have the authors made all data and (if applicable) computational code underlying the findings in their manuscript fully available?**

Reviewer #1: **No: **The data and code are said to be made available upon publication.

Reviewer #2: None

PLOS authors have the option to publish the peer review history of their article (what does this mean?). If published, this will include your full peer review and any attached files.

Reviewer #1: **Yes: **Florent Meyniel

Reviewer #2: No
---

## [Decision Letter · Decision Letter 1]

27 Jul 2024

Dear Dr. Murawski,

Thank you very much for submitting your manuscript "The neural dynamics associated with computational complexity" for consideration at PLOS Computational Biology. As with all papers reviewed by the journal, your manuscript was reviewed by members of the editorial board and by several independent reviewers. The reviewers appreciated the attention to an important topic. Based on the reviews, we are likely to accept this manuscript for publication, providing that you modify the manuscript according to the review recommendations.

Sincerely,

Daniele Marinazzo

Section Editor

PLOS Computational Biology

Daniele Marinazzo

Section Editor

PLOS Computational Biology

Reviewer's Responses to Questions

**Comments to the Authors: **

Reviewer #1: I thank the authors for their response which address my comments.

I would recommend that they screen their paper for acronyms that are not defined on their first occurrence (if defined at all). For instance, a number of brain regions are not defined beyond their acronyms: CON, dACC, IPS, etc.

Reviewer #2: The authors have done a good job of addressing my concerns about the previous version of the manuscript. However, I still have a few remaining concerns. The importance of these issues hovers somewhere between major and minor.

1. The authors state that they cannot analyze RT because the task is not self-paced. I do not follow this logic: RTs are typically analyzed in speeded reaction time tasks that have a maximum response time threshold. So why not look at them? The reason that this is important is because RTs provide an independent measure of trial difficulty. 

2. Although the terminology and task parameters are much clearer now, I still don’t fully grok the basic idea. On the one hand, proof hardness characterizes the complexity of verifying that a solution to a problem is correct. Unsatisfiable problems have high proof hardness, which because they require verifying that no witnesses exist, might require more than polynomial time. Therefore these trials should be difficult. On the other hand, unsatisfiable problems with low typical case complexity can be solved by applying logical rules. In other words, these trials are easy. So which is it–easy or difficult? I guess the operating word here is “might” (require more than polynomial time). I really believe that the paper will more impactful if the authors address this issue concretely.

3. Relatedly, the authors write in the cover letter (point 4) that they provided an example of the task on lines 139-148, but I can’t find any example at these lines.

**Have the authors made all data and (if applicable) computational code underlying the findings in their manuscript fully available?**

Reviewer #1: **No: **The data and code are not currently available, but the authors include an OSF project with currently empty folder, which they say will host the data and code upon acceptance.

Reviewer #2: Yes

PLOS authors have the option to publish the peer review history of their article (what does this mean?). If published, this will include your full peer review and any attached files.

Reviewer #1: No

Reviewer #2: No

Figure Files:

Data Requirements:

Reproducibility:

References:

---

## [Decision Letter · Decision Letter 2]

30 Aug 2024

Dear Dr. Murawski,

We are pleased to inform you that your manuscript 'The neural dynamics associated with computational complexity' has been provisionally accepted for publication in PLOS Computational Biology.

Best regards,

Daniele Marinazzo

Section Editor

PLOS Computational Biology

Daniele Marinazzo

Section Editor

PLOS Computational Biology

Reviewer's Responses to Questions

**Comments to the Authors: **

Reviewer #2: The authors have addressed all of my remaining concerns about the manuscript.

**Have the authors made all data and (if applicable) computational code underlying the findings in their manuscript fully available?**

Reviewer #2: Yes

PLOS authors have the option to publish the peer review history of their article (what does this mean?). If published, this will include your full peer review and any attached files.

Reviewer #2: No

---

## [Editor Report · Acceptance letter]

16 Sep 2024

PCOMPBIOL-D-23-01530R2 

The neural dynamics associated with computational complexity

Dear Dr Murawski,

I am pleased to inform you that your manuscript has been formally accepted for publication in PLOS Computational Biology. Your manuscript is now with our production department and you will be notified of the publication date in due course.

With kind regards,

Zsofia Freund
